# Pain Perceptions, Suffering and Pain Behaviours of Professional and Pre-Professional Dancers towards Pain and Injury: A Qualitative Review

**DOI:** 10.3390/bs13030268

**Published:** 2023-03-17

**Authors:** Andrew Soundy, Ja Yee Lim

**Affiliations:** School of Sport and Exercise Rehabilitation Sciences, University of Birmingham, Birmingham B15 2TT, UK

**Keywords:** dance, pain, injury, career outcomes, performance, behaviours, perceptions, attitudes, review, qualitative

## Abstract

Limited data exist that describe the experiences of pain and injury in dancers. The purpose of the current study was to understand pain perceptions, suffering and pain behaviours associated with pre-professional and professional dancers and to consider the psychosocial factors that influence suffering, behaviour and perceptions of pain. A thematic synthesis review was undertaken in three stages: (1) A systematic search using pre-defined search terms was conducted until 17 November 2022. Qualitative studies were included if they captured dancers’ perceptions, experiences, or the behaviour of dancers towards pain and injury. (2) Quality appraisal and certainty assessments was performed. (3) A five-phase synthesis generated themes that included a certainty assessment score. Twelve studies with 290 dancers met the inclusion criteria. The aggregated mean age was identified as 28.5 years. No studies were excluded following the quality appraisal stage. Nine studies included professional ballet dancers. Three themes were developed: (1) developing positive and adapted perceptions and behaviours towards pain, (2) the impact and danger of embracing pain and the risk of injury and (3) factors that influenced the response to injury and the ability to perform. This review has highlighted the experiences of pre-professional and professional dancers towards pain and injury. Practical implications for healthcare professionals, employers, choreographers and dancers are provided. Further research is required given the limited evidence base.

## 1. Introduction

Dance is perceived as a form of art that revolves around evoking an emotional response from an audience and creating displays of aesthetic beauty [1]. Pursuing dance as a professional career is extremely physically demanding. The daily routine of rehearsals, classes and performances force dancers to work themselves to the extreme, often pushing the boundaries of ‘the principles of human design’ [2]. One result of this is a high level of injuries [3]. Review evidence [4] identifies a point prevalence of professional dance injuries between 48% and 74%, with an annual rate of between 55% and 90% [5]. Most injuries are sprains, strains, tendinopathies and stress fractures of the back or lower extremities. Severe injury is a factor that influences the decision to retire, and most professional dancers retire in their thirties [6]. Research is needed that further understands the impact of injury. This includes understanding the time lost from work or activities, the cost of care, if any loss is temporary or permanent and the psychological impact of injury [7]. The psychological impact of injury is particularly important. Professional dancers often experience significant psychological distress, anxiety and bewilderment. This significantly influences their ability to perform and their career [2,8].

Unfortunately, pain is almost inevitable in dancers’ lives. Pain plays a critical role in performance and can significantly impact the career of dancers [9]. Pain is complex and needs to be defined. The International Association for the Study of Pain (IASP; https://www.iasp-pain.org/ accessed on 9 January 2023) defines pain as “an unpleasant sensory and emotional experience associated with, or resembling that associated with, actual or potential tissue damage.” The components of pain include nociception (detection of tissue damage by fibres), perception of pain (understanding of pain triggered by noxious stimulus or lesions, but can occur without nociception), suffering (negative psycho-emotional responses that result from the experience of pain) and pain behaviours (actions or activities that result from pain and suffering). This paper will focus on pain perceptions, suffering and pain behaviours, which illustrate how injury is managed.

Understanding the experience of pain for professional and pre-professional dancers is complex. For instance, professional dancers can perceive pain as a positive experience [6]. The relationship between injury, pain perceptions, suffering and behaviours must be understood within the culture of dance. For instance, dancers may fear stigma or negative reactions (including losing their contract) from employers if an injury is discovered [8,10]. Reviews of the current evidence are one way clarifying the complex relationship between the components of pain and injury. Within the existing literature, past reviews have identified that some psychosocial and psychological factors negatively impact injury incidence and performance outcomes [8,11]. Review evidence has also briefly discussed how performance and progression are associated with the perceptions of pain, suffering and behaviours [11]. However, to the best of the authors’ knowledge, no past reviews have provided a detailed consideration to the context and meaning of injury expressed by dancers. A review of qualitative studies would allow a more comprehensive and in-depth understanding of the experiences of pain and injury in dancers [12,13]. Qualitative articles focusing on pain in professional and pre-professional dancers are ideally placed to reveal this complex relationship [1]. For instance, qualitative research has identified that pain is not just interpreted as a bad or negative experience but as integral to performance and the dancer’s career [14].

The complex nature of pain requires professional dancers to be psychologically adaptive and generate coping mechanisms [6]. Without this, there is a danger that a dancer’s career and performance could be jeopardised [2]. A qualitative review would be able to establish the complex lived experiences of dancers. Such findings could aid the effectiveness of pain management by health care professionals, and this could lead to enhanced performance and career continuity [15]. It is important to note that health care professionals lack understanding and empathy of the pain experiences of dancers, and this misunderstanding often results in the mismanagement of the injuries [1,16].

Given the above, the aim of this qualitative review was to: (a) gain a deep understanding of pain perceptions, suffering and pain behaviours associated with pre-professional and professional dancers and (b) identify the psychosocial factors that influence suffering, behaviour and perceptions of pain.

## 2. Material and Methods

A thematic synthesis approach [17] in three stages was undertaken for this review. The ENTREQ framework [18] was used to aid reporting of this section. A subtle realist ontological world view was adopted for this review as the authors looked to focus on the common experiences of dancers relating to pain and injury [19]. The following three stages were undertaken:

### 2.1. Stage 1: Systematic Search

The search processes were undertaken using the PRISMA guidelines [20].

#### 2.1.1. Inclusion Criteria

Inclusion criteria were based on the SPIDER acronym [21] (Sample, Phenomena of Interest, Design, Evaluation, Research type):

Sample: Dancers over the age of 16 years were included. Dancers generally become professional at this age [22]. Studies were included if they included dancers that trained full time regardless of whether they were identified as pre-professional (those who are in full time training, or those who have advanced training and commit to full time training, but may earn a living by other means). Studies that had mixed populations could be included if distinct results on dancers were provided.

Phenomena of Interest: Studies were included if they had identified perceptions and experiences of pain, suffering or pain behaviours and identified how these aspects impact performance or career outcomes of dancers. Studies were also required to include coping and management strategies and factors which influence decisions. Qualitative research is ideally suited to this [23]. Professional groups as well as pre-professional groups were included, with a focus on the most common experiences, perceptions and behaviours. Where articles focused on injury/ies, they had to include a focus on the experience, perception or behaviour related to pain as well. Multiple articles generated from authors that used the same the dataset were only included if both review authors agreed that the results were distinctly different within each article.

Design: Any type of qualitative methodology was acceptable to document the experiences of pain. Examples could include variations of grounded theory, phenomenology, ethnography and participatory approaches. Systematic reviews, meta-synthesis, reflection pieces, commentaries, theses, books and studies that were observational, quantitative and mixed methods were excluded.

Evaluation: Data collection of any form capturing qualitative experiences and perceptions relating to pain, suffering and injury in professional or pre-professional dancers were included. This could include interviews, field diaries, qualitative observations or other methods that documented experiences or perception.

Research Type: Only qualitative designs were included.

#### 2.1.2. Data Sources

Electronic databases were searched systematically from inception until 17 November 2022. Four electronic databases were searched via EBSCOhost core collection: PubMed, CINAHLplus, Sport Discus and PEDro. Following this, the first 20 pages of the electronic search engines Google Scholar and ScienceDirect were searched. The titles of all identified articles were screened. Hand-searching reference lists of included and related articles was undertaken to ensure that the existing dance literature surrounding pain experiences and perspectives were thoroughly searched [24]. An article was included when it was considered that it satisfied the eligibility criteria based on the SPIDER acronym [21].

#### 2.1.3. Electronic Search Strategy

Searching with key terms and their synonyms was undertaken. Key terms included: (dance or dancers or danc *) AND (perspectives or perceptions or experiences or reflections or behaviour) AND (pain or injury or trauma) OR (managing or rehabilitation or coping) AND (qualitative or exploratory).

#### 2.1.4. Study Screening Methods and Data Extraction

The primary author read the title and abstract of the article for potential inclusion. If unsure, the author would read into the methods, discussion and results sections to ensure articles that fit the eligibility criteria were included. A standardised data extraction table was used to capture study findings.

### 2.2. Stage 2: Quality Appraisal

The primary author used the modified COREQ 13-item checklist [25] adapted from Tong et al. [26]. The checklist comprised three domains: (1) research team and reflexivity, (2) study design and (3) analysis and findings. A quality assessment was undertaken to assess the quality of the included studies, exclude studies and identify any fatally flawed studies [27,28]. Studies were identified as fatally flawed if: (a) the reporting was limited or non-existent, (b) the results reported were limited or identified as not representing the interview schedule and processes undertaken or (c) the COREQ score was below 6 and further discussions identified a critical weakness based on three criteria. The three criteria included (a) how well developed the area of study was by considering the development of the interview schedule and the results, (b) if the methodological processes identified explained how the results were obtained and if they seemed believable and (c) if the findings support the common findings from other studies.

### 2.3. Stage 3: Synthesis

Thematic synthesis was undertaken in five phases. A full audit trail of the synthesis process can be obtained from the primary author. Phase 1 involved open coding [29] by the primary author, and quotes of interest were identified. Phase 2 involved developing descriptive themes using axial coding and mind-mapping. Axial coding was undertaken to reorganise and determine dominant ideas among the codes [30]. Common themes and sub-themes were extracted from the codes for the results section [31]. Phase 3 involved further categorising the quotes assigned under each theme into sub-themes and codes. Phase 4 identified clear associations between the themes by using idea webbing [19]. Phase 5 involved considering data saturation and presentation of themes as well as applying the certainty assessment [32]. The presentation of the results was adapted by the corresponding author for the purposes of publication.

## 3. Results

A total of 1401 studies were screened of which 12 studies met the eligibility criteria. Figure 1 presents the PRISMA flow diagram detailing the full search process.

### 3.1. Demographics

Two hundred and ninety people were identified (35 females and 18 males, 237 unknown) across the 12 studies [1,2,6,14,16,33,34,35,36,37,38]. All studies included ballet dancers except four studies. The study conducted by Harrison and Ruddock-Hudson [6] included 8/20 ballet dancers and 5/20 contemporary dancers. The remaining dance categories were ballroom (n = 1), salsa (n = 1), performing arts (n = 2), commercial dancing (n = 1), Broadway (n = 1) and cruise ship dancer (n = 1). Markula [16] included 14 contemporary dancers. Both studies by Tarr and Thomas [35,36] did not provide a breakdown of categories but state “the research focused largely on contemporary, independent dance sector rather than ballet” [36]. The professional dancers’ experience ranged from 1 to 20 years. The dancers’ ages ranged from 21 to 45 years. The aggregated mean age of the dancers was 28.5 years from three studies reporting means [1,6,34]. The standard deviation could not be calculated due to missing data. The other studies did not detail professional dance experience and the age of the interviewed dancers. Six studies came from the UK [14,23,35,36,37,38], two from Canada [1,16], one from Australia [6] and three from the Netherlands [2,33,34]. The same data set was used in the studies by Wainwright and Turner [14,37], Wainwright et al. [38] and Tarr and Thomas [35,36]. Table 1 provides a summary the demographic details of the included studies. 

### 3.2. Critical Appraisal

Within this study, critical appraisal found that two studies were identified as high-quality, scoring 10/13 [23] and 8/13 [25]. The other studies were identified as poorer quality, scoring 6/13 or less. The critical appraisal identified that for domain 1, which considered details of reflexivity and research, the average score was 2.1/5. For domain 2, which considered study design, the average score was 1.1/5. All 12 studies did not report any non-participation or gave details of data saturation. Domain 3, which considered analysis and findings, provided an average score of 2.3/5. The clarity of minor themes was not given in three studies [14,16,26]. All studies scoring six or less on the COREQ were assessed by both authors. The authors identified that across all studies there was sufficient detail given to be included in the review. The quality appraisal was integrated into the certainty assessment score.

### 3.3. Synthesis

Three main themes were identified. These included (1) developing positive and adapted perceptions and behaviours towards pain, (2) the impact and danger of embracing pain and the risk of injury and (3) factors that influenced the response to injury and the ability to perform.

## 4. Theme 1: Developing Positive and Adapted Perceptions and Behaviours Relating to Pain

This theme identified how dancers had come to understand and experience pain. It also identified how cognitive strategies were used to cope with pain and enhance performance. This theme was made up of three sub-themes, including (1) the perception of pain, (2) the experience of pain and (3) the cognitive strategies used to enhance performance and cope with pain.

### 4.1. Sub-Theme 1: The Perception of Pain (Moderate CerQual Evidence Rating)

All studies reported that dancing at a high level was associated with the experience of pain. Pain was described by many dancers as inevitable and always present [1,2,6,16,23]. Therefore, the acceptance of pain was perceived to be unavoidable, ever-present [1,2,6,33] and normalised [1,6,14,16,34,38]. Some could acknowledge pain as damaging but a part of being a professional dancer [1,16,23]. Accepting pain was part of the calling to be a dancer or could be embraced because of their love for dance [1,14], but also because it was part and parcel of performing [14] and was required to achieve career goals [1].

The idea of having an ideal body includes the requirement that it can function well [33], and injury could be considered as the point of not being able to move as the dancer would like. A remark from the group discussions in Bolling et al., [34] was that “pain doesn’t mean injury”. Tarr and Thomas [35] further identified that injury was not so much associated with the presence of pain but with the limits of the body. The dancers understood that if you want to be good and succeed, you have to suffer. This was created through an understanding of which expressions of pain were acceptable during training, for instance, crying could be forbidden within the culture [33]. Further to this, pain could be considered as ‘good’ pain, planned pain or in a sense something you do to yourself (e.g., brought on by strengthening or training, vibrating muscles, cramp, tolerable), and as ‘bad’ pain, where there is a need to stop (e.g., unexpected, sharp, shooting, injury pain, unexpected, burning or fire, not disappearing, or constant or unbearable) [35,36].

### 4.2. Sub-Theme 2: The Experience of Pain (Low CerQual Evidence Rating)

The experience of pain could provide dancers with a feeling and indication that they were working hard, and this experience would help them become better [2,6,14,16]. For instance, one dancer identified that she was not working hard if she did not experience pain when dancing [16]. Indeed, dancers identified pain as an optimistic feeling [1,2,16]. One dancer stated that pain could be felt as pleasurable and provided a sense of improvement [1]. This should be understood within the mindset that the right kind of training, will power and exercise can create a body which will perform. This was reinforced by rewards, which could create an understanding that pain was part of that reward. For instance, an individual in the study by Aalten [33] stated “every night I sat down in front of my bed and pushed my legs wider until they really hurt. The next day I could go a bit further” (pp. 60–61).

### 4.3. Sub-Theme 3: Cognitive Strategies Used to Enhance Performance and Cope with Pain (Moderate CerQual Evidence Rating)

Dancers perceived a great need to remain competitive among their peers [1,6,14,38]. Dancers would do anything to remain employable and viable members of their subculture. Several cognitive strategies were employed to enhance continued performance, including the following: (a) Adapting the way they dance to avoid aggravating the injury. If this was achieved, some would not perceive themselves as injured [6]. (b) Reframing the perception of pain to adapt the pain behaviour and regarding the pain experience as an opportunity to enhance a dancer’s technique [6,23,35,38]. Alternatively, understanding that what once was understood as pain is no longer seen as pain. As one participant from the Aalten [33] study identified, “I discovered that it is possible to focus yourself mentally up to a point where you do not feel the pain anymore the moment you get on stage. In a way you surpass the pain” (pp. 63). (c) Thought stopping was used to prevent the dancer from dwelling on the negative impact of an injury that has prevented performance [25]. (d) Being willing to ‘listen’ to their body after a severe injury [1] as well as being able to map verbally or textually the experiences [35]. Dancers could discover self-awareness, which was important for preventing further injuries [1,2,23]. Dancers learnt the early warning signs of injury [1] and developed an enhanced ability to gauge their body’s limits [23]. (e) The experiences of pain could enhance artistic sensitivity and produce psychological growth as a professional. This process helped the dancer to be more grateful for the ability to return to dance after injury and enjoy sharing expressions and feelings through dance, rather than be so concerned about their technical performance [38]. (f) Identifying the importance of body care upon reflection on their injury experience and their existing approach to body care [1,2,23]. For a dancer, this was to work out the limits of the body instead of an ‘ideal’ needed for performance [33]. (g) Risk-taking was an important and calculated choice, and included risking pain and injury by going to their bodily limits or beyond their limits [1,6,23,38]. For instance, pain was embraced to achieve their personal goals [1]. Risk taking also included masking the experience of pain and injury from people who could influence their career, as dancers feared that their career opportunities would be revoked [1,23]. For instance, one dancer actively chose to dance through the experience of pain to achieve her personal career goals [1].

## 5. Theme 2: The Impact and Danger of Embracing Pain and the Risk of Injury

This theme identified the abilities of the dancers to recognise and manage injuries and the consequences if this did not happen. Two sub-themes were identified in this theme, including (1) the ability to recognise and manage pain and injury and (2) the cost of being out of performance.

### 5.1. Sub-Theme 1: The Ability to Recognise and Manage Pain and Injury (Low CerQual Evidence Rating)

The process of embracing pain and understanding the impact and context for injury could represent a difficulty for dancers. The inability to recognise serious injuries could jeopardise their career or performance [2,6,23,35]. For instance, it caused one dancer to rupture her tendons, which put her out of performance [2]. Related to this, some dancers who managed their pain inappropriately also risked being out of performance. They included hiding their pain and injuries, dancing through them [1,6,33] and being in self-denial about their injury [38]. These actions would prolong their injuries and risk their career prospects [1,6,38]. Further to this, injuries could be performance-limiting [2,6,16,37,38].

### 5.2. Sub-Theme 2: The Cost of Being out of Performance (Low CerQual Evidence Rating)

Being out of performance due to pain or injury could be perceived as a personal failure. For instance, one dancer suffered an injury which prevented her from performing for nine months [37]. An injury can also affect the career progression of dancers [6,35,37,38], where it impinges on their career and employment opportunities [6]. Dancers felt liable for losing precious time out of their short careers [37]. This perception led to a sense of guilt and responsibility among dancers about their own behaviours towards pain [1,6,16,37]. It could also lead to a loss of identity, negatively affecting a dancer’s performance or career [1,38]. Such experiences could force a dancer to quit performing [1] and affect a dancer’s confidence in returning to performance life [37].

## 6. Theme 3: Factors That Influenced the Response to Injury and Ability to Perform

This theme considers the main psychosocial factors that impact the perceptions of pain and ability to manage pain. Five sub-themes were identified as impacting the response to injuries and ability to perform. These included the following:

### 6.1. Sub-Theme 1: The Ability to Self-Manage (Moderate CerQual Evidence Rating)

Several self-management strategies used by dancers were identified. These included: (a) Adapting the way they dance without aggravating the injury. If they were capable of doing this, they would not perceive themselves as injured [6] because the injury could be worked through [34]. (b) Tolerating the pain from injuries [23,33,38] and understanding that an injured dancer often can and will dance through pain [34]. This could include dancing through nagging pain or ‘niggles’ and being alert to an ‘active call’ or using breaks in the schedule to rest. This was created through experience and self-reflective awareness [35]. Some dancers could not self-manage their pain and injury, consequently affecting continued performance throughout their career. This was due to the negative psychological impact that the experience of pain and injury had [6,16,38]. It affected their self-confidence in dancing [6,38].

Self-management could go wrong for several reasons, including the following: (a) Neglecting body care. This meant taking their injuries lightly by ignoring them and continuing to dance [1,6,16,38]. (b) A late realisation of one’s body limit [2,6]. Both factors could cause some dancers to exit their careers prematurely [1,6].

### 6.2. Sub-Theme 2: Pain Threshold and the Culture of Hardiness (Moderate CerQual Evidence)

Dancers had varying pain thresholds. Some dancers were able to continue dancing unaffected by injuries, whereas the same injuries stopped others from dancing [34,38]. However, there was a belief that dancers should persist through pain and injury [2,16,34,38]. Only severe injuries were considered valid for rest [1,16]. For instance, one dancer stated, “a dancer should ‘suck it up’ unless he...is ‘really injured’, which she went on to define in terms of... ‘requiring surgery’.” (p. 159) [1]. Understanding overload and body limits was essential to this [34]. The expression of pain was not well-received by teachers or employers [2,33]. For instance, a dancer stated “Crying if you were in pain was absolutely forbidden...the reaction was always: ‘Keep smiling! You are the one who wants to be here, if you don’t like it you can leave.” (p. 117) [2]. Alternatively, having the ‘mental toughness’ to dance through injury “distinguished those who became professional *dancers*” (p. 59) [38].

### 6.3. Sub-Theme 3: Fear-Related Preconceptions (Low CerQual Evidence)

The fear of missing out on career opportunities [1,6,23,38] played a significant role among dancers. Reasons for this included: (a) The belief that they might lose opportunities to perform or might jeopardise their career if they reported their injury to their employers [1,16,23,38]. Some dancers felt that employers were uncertain about giving coveted roles to dancers they knew were injured [23]. (b) Dancers perceived that they would be judged as weak or lacking commitment and passion for the profession if they admitted to having an injury [1,6,38].

### 6.4. Sub-Theme 4: Performance Pressures (Moderate CerQual Evidence)

External pressures within the dance industry could act as a dancer’s motivation to dance through injury [2,23,33,38]. For instance, dancers would dance through pain and injury to perform for a scheduled show, even though rest was necessary for their injuries. Reasons for this included: (a) A perceived responsibility for their role and considering their role as irreplaceable in a choreography [2,38]; this could be due to the limited dancers suitable to dance their role in a choreography [2]. (b) The perception that there was no choice but to continue [1,38]. For instance, a dancer may have no substitutes, so they have to perform [1]. (c) Dancers did not want to disappoint others [1,2,16,33]. For instance, peers who were performing in the same choreography as them [2]. Similarly, pre-professional dancers felt that they were disappointing their teachers if they told their teachers about their pain and injury [1]. Dancers were anxious about approaching their teachers and employers concerning injuries as they could ‘make or break’ their dancing career. (d) Dancing was purely fulfilling and addictive [37,38].

## 7. Discussion

To the best of the authors’ knowledge, this is the first review that considers the importance of suffering as a part of dance, and identifies factors influencing suffering, pain behaviours and pain perceptions. Past reviews have identified and discussed the psychosocial factors influencing pain perceptions and behaviours towards injury [8,39]. However, the past literature has never brought these findings together to help understand how they have shaped perceptions, behaviours and suffering that could encourage or jeopardise a dancer’s career.

One of the most important pain behaviours identified in the current review was the dancer’s recognition of their own body limits. This included self-awareness and identification of early warning signs, understanding the physical limits of the body and learning to listen to the body after severe injury. Dancers would adapt the way they danced to avoid injury or regard injury as an opportunity to change and enhance technique. Future research needs to establish interventions that enable coping and continued success. Past studies support this as they highlight that good self-awareness increased the chances of career success [40,41]. Dancers identified that being able to take risks with suffering and injury was important, and this resulted in working at the limit of the body, which could produce meaningful goals and the continuation of their career. Future research would benefit from understanding the concept of risk further. Being able to establish goals is regarded as important for career success [36]. Past research [1] has identified that enduring pain would enhance a dancer’s worth to employers. Considering the influence employers have on the dancers’ careers is important for future research.

The current review revealed that suffering could be offset against the importance of the following considerations: First, a dancer needs to understand the limits of their body before injury and that pain can be ‘good’ in situations that are ‘controlled’ although present a risk. Second, positive outcomes can result from pain and illustrate that the dancer is working sufficiently hard. This appeared to be associated with optimism for the future and dancers considered this necessary for progress [42]. However, research is needed to understand how dancers are capable of perceiving pain in ways that facilitate continued performance throughout their careers.

Factors which influenced the career outcomes of dancers in the current review were psychosocial or sociocultural in nature. Dancers who tended to push themselves further and masked pain from their colleagues and employers perceived a positive impact on their performance and career opportunities. This finding is supported by past research [43,44]. A recent review by Mainwaring and Finney [8] suggested that there were implications of injuries on dance companies’ finances such as the loss of ticket sales and concurrently having to pay injured dancers who could not perform. This may be used to explain the existing sociocultural unacceptability of pain and injury within the dance industry. It may also explain the common behaviour of masking pain and embracing suffering. Further exploration is needed to understand how this competitiveness influences a dancer’s pain behaviours.

The current findings identified that the (in) ability to embrace the culture of pain and possess higher pain thresholds, could influence a dancer’s performance and career opportunities for pre-professional and professional dancers. This is supported by the past literature [43,44]. Furthermore, performing through pain was often glorified and dancers were regarded as having mental toughness [45]. In agreement with past dance reviews and studies, the negative psychological impact of pain increased injury susceptibility and recurrence, consequently causing dancers to miss training and increase the risk of losing their careers [8,46]. Past research has also suggested that a dancer’s social identity is connected to their profession, hence being unable to dance is a major threat to their identity [11,38]. The current findings support this, and further research is necessary to explore how psychological adjustment, a sense of social identity or self-efficacy influence continued performance. Further study is also needed to identify why some dancers are not able to overcome the psychological impact.

Professional dancers in the current review found it difficult to distinguish performance pain from pain that meant an injury was present. One reason for this is the consistent state of pain experienced by dancers [5,47]. Further studies can explore why some dancers are able or unable to distinguish performance and injury pain and if having this ability would cultivate better pain behaviours towards injury. Self-awareness and self-management skills for injuries are critical factors for increased career success [40,41] and can facilitate a pathway towards continued performance. Further studies should explore the factors that may facilitate this transition. Inappropriate strategies of pain management, such blocking out pain, require further research.

Past studies have found that dancers with the ability to distinguish pain [6], who have good self-awareness and self-management skills, have increased chances of career success [40,41]. This may explain why self-neglect and a lack of self-awareness and attention to the body’s physical limits appeared to jeopardise a dancer’s career and performance in the current review. This also appeared to create a sense of guilt and responsibility in dancers. Future research should investigate if this sense of guilt and perceived responsibility may help with to change a dancer’s pain behaviours for better injury management throughout their career.

### 7.1. Practical Implications

The current results have important considerations for enhancing interactions and rehabilitation with dancers. For instance, employers and choreographers work closely with dancers and have a major influence on their career opportunities and performance [1]. Because of this relationship, dancers could benefit from a safe space to reveal injuries or pain behaviours that are being managed, and this would mitigate risk and improve the management of an injury. Related to this, dancers would benefit from education and mentoring about what the limits of the body are and examples of what occurs if the limit is ignored. Dancers would benefit from understanding what future performance and roles are, and this would allow time for planning how to mitigate injury risk in a more specific way. When injuries occur, psychological assessments may be beneficial to understanding the adjustment following injury and identifying if the suffering experienced could be detrimental to the dancer’s career. Self-efficacy following injury could be assessed as this construct is appears to be directly affected in the current results. Dancers would likely benefit from understanding the current findings. In particular, this includes cognitive strategies or pain behaviours that enhance their ability to cope with their career and performance demands.

### 7.2. Limitations

Several limitations are acknowledged: (a) Results from the included studies were retrospective in nature and given from the dancers’ perspectives. Understanding the perspective of others, including family and professional staff, would be useful. (b) Most of the included studies focused on ballet dancers and recommendations should be taken with caution when generalising findings [8]. (c) Only articles published in English and peer-reviewed qualitative articles were included. Results in books, theses and unpublished work were excluded. (d) One author was involved in screening and including studies for this review. Hence, the consistency of the screening and inclusion process might have been affected as relevant studies might have been missed [48]. (e) Lastly, pre-professional dancers were included due to the limited nature of the dance literature. The contributing studies, however, may make the results more appropriate for current professional dancers.

## 8. Conclusions

The present qualitative review has highlighted the significance of the experience and perception of pain, the ability to manage pain as well as the psychosocial factors which influence this ability. Cognitive strategies were essential to managing pain and benefited from experience. Further research is required to determine the benefits and effectiveness of using the above findings to benefit dancers.

## Figures and Tables

**Figure 1 behavsci-13-00268-f001:**
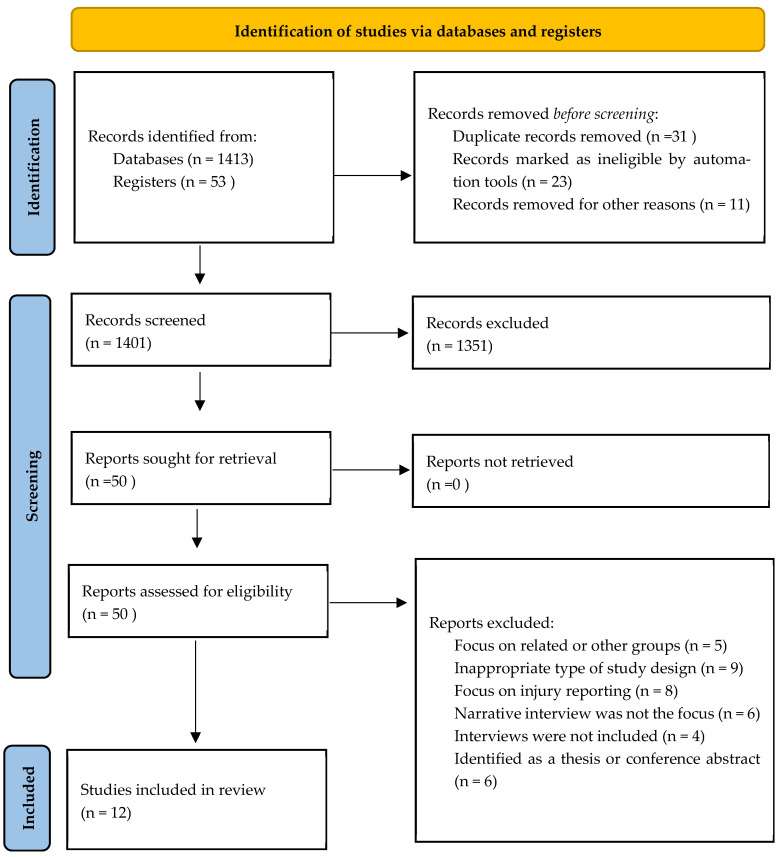
The 2020 PRISMA follow diagram (Page et al., 2020) [20].

**Table 1 behavsci-13-00268-t001:** The demographic details of included studies.

Study	Participants, Sampling, Setting and Aim	Methodology, Methods and Analysis
Aalten [2]	*Participants*: 25 female professional ballet dancers. 9 dance students at professional ballet schools. Male and Female students (numbers not identified). *Age*: Not detailed. *Sampling*: Snowball. *Setting*: Public meeting places or at the participants home. *Aim*: To analyse the meaning of injuries and pain in the context of ballet culture. *Geographical location*: The Netherlands.	*Methodology*: Ethnography. *Outcome measures*: Formal and informal interviews. *Observations* Biographical interviews. *Analysis*: Phenomenological approach.
Aalten [33]	*Participants*: 25 female professional ballet dancers. 9 dance students at professional ballet schools (no gender breakdown given). *Age*: Not detailed. *Sampling*: Snowball. *Setting*: Public meeting places or at the participants home. *Aim*: To analyse the meaning of injuries and pain in the context of ballet, drawing on the understanding of social identity and how the body is understood by the dancers. *Geographical location*: The Netherlands.	*Methodology*: Ethnography. *Outcome measures*: Formal and informal interviews. Observations. Biographical interviews. Autobiographies of dancers. *Analysis*: Sociological approach focused on the body.
Bolling et al. [34]	*Participants*: 10 professional ballet dancers (6 female, 4 male). *Age*: Average 27 years (range 20–35 years) *Sampling*: Maximum variation. *Setting*: Not identified. *Aim*: To consider the dancers’ perceptions of injury. *Geographical location*: The Netherlands.	*Methodology*: Grounded theory. *Outcome measures*: Focus groups. *Analysis*: Grounded theory analysis.
Harrison & Ruddock-Hudson [6]	*Participants*: 20 professional dancers (9 males, 11 female). *Age*: Average 35 years. *Sampling*: Convenience. *Setting*: Not identified. *Aim*: To explore the perceptions and experiences of injury, pain and retirement among professional dancers. *Geographical location*: Australia.	*Methodology*: Not identified (likely a type of phenomenology). *Outcome measures*: Semi-structured interview. *Analysis*: Content analysis.
Markula [16]	*Participants*: 14 contemporary dancers (no gender breakdown). *Age*: Range 18–30 years (no mean given). *Sampling*: Convenience. *Setting*: Not identified. *Aim*: To understand how contemporary dancers experience injuries. *Geographical location*: Canada.	*Methodology*: Deleuzian theoretical concepts used. *Outcome measures*: Formal face-to-face interviews. *Analysis*: Textual analysis.
Pollard-Smith & Thomson [23]	*Participants*: 8 ballet dancers (3 male, 5 female). *Sampling*: Snowball initially, followed by purposive and theoretical. *Setting*: No details. *Aim*: To explore the decision-making process of dancers who are seeking treatment for injuries. *Geographical location*: England.	*Methodology*: Grounded theory. *Outcome measures*: In-depth interviews. *Analysis*: Grounded theory analysis.
McEwen & Young [1]	*Participants*: 15 professional ballet dancers (13 female, 2 male). *Age*: Mean was 21 years. *Sampling*: Snowball. *Setting*: Selected by dancers’ public locations or homes of dancers. *Aim:* To explore how elements of risk-taking behaviours affect physical and emotional health of dancers. *Geographical location*: Canada.	*Methodology*: Unclear, likely a type of phenomenology. *Outcome measures*: Semi-structured interviews. *Analysis*: Thematic analysis.
Tarr & Thomas [35]	*Participants*: 205 (30 male, 175 female) 167 dancers (85 professional, 82 pre-professional (students). Gender breakdown of dancers not given. *Age*: 51% aged between 20–29 years. *Sampling*: Convenience and snowball. *Setting*: University. *Aim*: To consider the cultural and embodied experience of dancers in relation to pain. *Geographical location*: United Kingdom	*Methodology*: Cultural phenomenology. Outcome measures: Questionnaire. Semi-structured interviews. *Analysis*: Thematic analysis.
Tarr & Thomas [36]	*Participants*: 205 (30 male, 175 female). 85 professional, 82 pre-professional (students). *Age*: 51% aged between 20–29 years. *Sampling*: Convenience and snowball. *Setting*: University. Aim: 3 main aims: (1) identify how dancers distinguish between pain and injury and what this means for a cultural understanding of pain, (2) consideration of how visual representation of pain helps the understanding of injury and (3) how does movement style affect pain and injury. *Geographical location*: United Kingdom	*Methodology*: Cultural phenomenology. Outcome measures: Semi-structured interviews. *Analysis*: Thematic analysis.
Wainwright and Turner [14]	*Participants*: 22 professional dancers. *Age*: Not detailed. *Sampling*: Convenience. *Setting*: Royal Opera House and University. *Aim*: To understand interactions between injuries, dancers’ experiences of discomfort and social support. *Geographical location*: United Kingdom.	*Methodology*: Grounded theory. *Outcome measures*: Semi-structured interviews. *Analysis*: Grounded theory.
Wainwright et al. [38]	*Participants*: 22 professional dancers. *Age*: Not detailed. *Sampling*: Convenience. *Setting*: Royal Opera House and a University. *Aim:* Understanding the impact injuries can have on individual identities. *Geographical location:* United Kingdom.	*Methodology*: Unclear, likely grounded theory. Outcome measures: Semi-structured interviews. *Analysis*: Grounded theory.
Wainwright & Turner [37]	*Participants*: 22 professional dancers. *Age*: Not detailed. *Sampling*: Convenience. *Setting*: Royal Opera House and a University. *Aim*: Illuminate the embodiment of classical ballet. *Geographical location*: United Kingdom.	*Methodology*: Ethnographic. *Outcome measures*: Semi-structure interviews. Observations. *Analysis*: Thematic analysis.

## Data Availability

Data is available from the corresponding author.

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
