# Peer review of "Pain Perceptions, Suffering and Pain Behaviours of Professional and Pre-Professional Dancers towards Pain and Injury: A Qualitative Review"

_behavsci, 2023, doi:10.3390/bs13030268_

Round 1

Reviewer 1 Report

The study was mainly conducted on the analysis of statistical data, although it would be necessary to use interviews and conversations with respondents, since the dance environment is extremely sadomasachistic, people from this environment tolerate unimaginable pain and bring the body to injuries. Therefore, it would be necessary to pay attention in the study to the fact that the environment is cruel and toxic, and people from this environment need to be consulted by a psychoanalyst so that the attitude to pain and injury is adjusted adequately.

There are not enough analytical results in it, mainly a statement of statistical data is presented, without identifying the personal characteristics of the respondents of the dance environment.

Author Response

Author response: Thank you for your comments and time taken to review.

The study was mainly conducted on the analysis of statistical data, although it would be necessary to use interviews and conversations with respondents, since the dance environment is extremely sadomasachistic, people from this environment tolerate unimaginable pain and bring the body to injuries. Therefore, it would be necessary to pay attention in the study to the fact that the environment is cruel and toxic, and people from this environment need to be consulted by a psychoanalyst so that the attitude to pain and injury is adjusted adequately.

Author response: Thank you for these comments. This is a review of past studies  and mainly considers and focuses on experiences and summarise them. We believe we have updated the manuscript to ensure recommendations of psychological support are given.

There are not enough analytical results in it, mainly a statement of statistical data is presented, without identifying the personal characteristics of the respondents of the dance environment.

Author response: Thank you for these comments. Demographics of included studies are provided on page 6 between 241-250. We believe this provides a summary of characteristics. Limitation b identified the limits re having a lot of ballet dancers. We would be happy to add further limitations if you think it is required.

Reviewer 2 Report

Dear Authors,

Thanks!

Please:

Results

A total of 12 studies were eligible for inclusion.

------Error! Reference source not found. 202 ?

Please, justify.

Figure - PRISMA flow diagram detailing the full search process

Please, see Figure. Information is missing.

Material and Methods

Inclusion Criteria:

Sample: Dancers >16 years were included.

Please, insert mean and SD

Data Sources:

Electronic databases were searched systematically until 17/11/22. Four electronic da- 151 tabases were searched: PubMed, CINAHLplus, SportDiscus, PEDro. Following this, the 152 electronic search engine Google Scholar was searched. The titles of all identified articles 153 were screened.

Ok, but:

Web of Science Core Collection?

Kind Regards

Author Response

Author response. We have identified our responses to your comments below. We thank you for the time given to improve this manuscript.

Thanks!

Authors: Thank you for considering this manuscript.

Please:

Results

A total of 12 studies were eligible for inclusion.

Authors: This has been updated to identify the initial and final numbers.

------Error! Reference source not found. 202 ?

Please, justify.

Authors: We did a search for this but could not see it on the manuscript. Could you identify a line number where the problem exists. Thanks.

Figure - PRISMA flow diagram detailing the full search process

Please, see Figure. Information is missing.

Authors: Thank you we have made the boxes larger. This allows all information to be seen. 

Material and Methods

Inclusion Criteria:

Sample: Dancers >16 years were included.

Please, insert mean and SD

Authors: Thank you for this comment. We provide an aggregated mean of the studies on page 6 line 245.

Data Sources:

Electronic databases were searched systematically until 17/11/22. Four electronic da- 151 tabases were searched: PubMed, CINAHLplus, SportDiscus, PEDro. Following this, the 152 electronic search engine Google Scholar was searched. The titles of all identified articles 153 were screened.

Ok, but:

Web of Science Core Collection?

Authors: This was a core collection from EBSCOhost. We have updated this now and included this information. Thank you for your comments.

Reviewer 3 Report

- why did the authors remove authors' affiliations from the manuscript?

- Revise in title = ; to :

- What is the behavior of pain?

- Range date for data search should be clarified.

- SD for age should be reported.

- several instances of "e.g." which is not aligned with the citation style in MDPI.

- clarify what is: pre-professional dancers, professional dancers and ex-professional dancers, and semi-professional dancers.

- Related to the previous comment, it goes against the title and objective. Thus, it needs revisions.

- Why not use Scopus and WoS databases for search? the two most used in this type of research?

- Line 202: there is an error to revise.

- Figure 1: needs revision since as a reader, there is an error on why records were removed.

- In addition, phases (in blue) need revision.

- lines 241-244: please state which type of dancers.

- The flowchart does not follow the current PRISMA recommendations. Needs revision.

- How was pain measured in each study? please describe.

- Line 311: unclear sentence.

- lines 435: Do not confuse suffering with pain.

- A table with the 12 studies considered for this review is welcomed.

- Line 491: there is no reference to whether or not the participants are all professional dancers. Responding to the previous comment could help.

- lines 533-548: how can the authors propose clinical implications if they conducted a review? most importantly, why clinical, if the authors refer to practical suggestions?

- line 551: why stakeholders?

- line 555: why should grey science be considered?

- lines 559-560: this information goes against previous citations.

- overall, the objective of the study is unclear (type of dancers, professional type, etc), limitations of previous studies and the contribution of the present.

Author Response

Thank you for your comments. We have responded to each below. We believe the manuscript now is in a better shape. 

Comments and responses:

why did the authors remove authors' affiliations from the manuscript?

Author comment: One author is an editor for the journal and it was consider important that the reviewers were not influenced by this.

- Revise in title = ; to :

Author comment: This has been changed. Thanks.

- What is the behavior of pain?

Author: we have defined it within this section on page 2 line 58. “pain behaviours (actions or activities that result from pain and suffering or the expectation of it).

- Range date for data search should be clarified.

Author response: Thank you for this we have identified that the search was from database inception.

- SD for age should be reported.

Author response: We agree with the point. However, the aggregated mean was possible to calculate from studies. But without the full dataset for each study we were not able to calculate the SD.

- several instances of "e.g." which is not aligned with the citation style in MDPI.

Author response: thank you we have removed this style.

- clarify what is: pre-professional dancers, professional dancers and ex-professional dancers, and semi-professional dancers.

Author response: Thank you for this comment we have simplified the criteria to aid clarity with this..

- Related to the previous comment, it goes against the title and objective. Thus, it needs revisions.

Author response: Thank you for this observation. We agree and have changed the wording.

- Why not use Scopus and WoS databases for search? the two most used in this type of research?

Author response: In order to capture more articles and address this comment we searched the first 20 pages of www.sciencedirect.com. This only identified previously considered and excluded studies.

- Line 202: there is an error to revise.

Author response: We have changed the sentence here. Thank you.

- Figure 1: needs revision since as a reader, there is an error on why records were removed.

- In addition, phases (in blue) need revision.

Author response: Thank you we have considered the figure and updated it.

- lines 241-244: please state which type of dancers.

Authors: Thank you for this we have added types.

- The flowchart does not follow the current PRISMA recommendations. Needs revision.

Author response: We have updated the flow chart. I am not sure the additional features in the 2020 version are required. We would be happy to add if that is the case.

- How was pain measured in each study? please describe.

Author revision: We have clarified the inclusion criteria on page 3 which documents acceptable qualitative methods of data collection.

- Line 311: unclear sentence.

Author revision: This has been updated.

- lines 435: Do not confuse suffering with pain.

Author revision: thank you this reference has been updated.

- A table with the 12 studies considered for this review is welcomed.

Author revision: this has been added now to the manuscript.

- Line 491: there is no reference to whether or not the participants are all professional dancers. Responding to the previous comment could help.

Author revision: this has been detailed as much as possible now within the table and other areas.

- lines 533-548: how can the authors propose clinical implications if they conducted a review? most importantly, why clinical, if the authors refer to practical suggestions?

Author response: this has been changed to practical implications.

- line 551: why stakeholders?

Author response: This has been reworded.

- line 555: why should grey science be considered?

Author: it lacks peer review.

- lines 559-560: this information goes against previous citations.

Author response: this has been removed.

- overall, the objective of the study is unclear (type of dancers, professional type, etc), limitations of previous studies and the contribution of the present.

Author response: Thank you this has been made clear now. Thank you for your comments.

Reviewer 4 Report

Good morning.

I would like to congratulate the authors for the review presented. It is a quality work, with very interesting results, which perfectly synthesise the findings in the field of study analysed. 

In order to improve its final presentation, I propose a series of modifications.

Abstract.

The information on the results and, especially, the conclusions of the study should be improved.

Introduction.

In the introduction, the background of the problem is clearly exposed.

The general definition of pain and the components of pain exposed (page 2) need to be referenced, with their corresponding citations in the references section.

Page 2. Line 64: "perceivje".

Page 2. Line 77: clarify what is "PDs". It is not previously specified what this abbreviation refers to. It appears continuously in the following sections.

Page 2. Avoid continually repeating "e.g.".

Page 2. Line 101. "The aims".

Methods.

Methods are described in detail.

Results.

Page 5. Line 202. “Error! Reference source not found.”

Page 5. Unable to read Figure 1, some words and phrases cannot be read.

Page 7. line 317. Change to "Sub-theme 3: Cognitive..."

Discussion.

Expand with future lines of research.

Conclusion.

This is a very limited section. The main findings of the study should be cited.

References

The references section is not correct. References should be adapted to the journal's standards.

Author Response

Author 4

Comments and Suggestions for Authors

Good morning.

I would like to congratulate the authors for the review presented. It is a quality work, with very interesting results, which perfectly synthesise the findings in the field of study analysed. 

Author response: Thank you for this feedback

In order to improve its final presentation, I propose a series of modifications.

Author response: Thank you for taking the time to provide this information.

Abstract.

The information on the results and, especially, the conclusions of the study should be improved.

Author response: This information has been updated thank you.

Introduction.

In the introduction, the background of the problem is clearly exposed.

The general definition of pain and the components of pain exposed (page 2) need to be referenced, with their corresponding citations in the references section.

Author response: Thank you we have updated this.

Page 2. Line 64: "perceivje".

Author response: this has been updated.

Page 2. Line 77: clarify what is "PDs". It is not previously specified what this abbreviation refers to. It appears continuously in the following sections.

Author response: This has been updated

Page 2. Avoid continually repeating "e.g.".

Author response: These have been removed.

Page 2. Line 101. "The aims".

Author response: this has been updated.

Methods.

Methods are described in detail.

Results.

Page 5. Line 202. “Error! Reference source not found.”

Author response: This has been updated.

Page 5. Unable to read Figure 1, some words and phrases cannot be read.

Author response: This has been updated.

Page 7. line 317. Change to "Sub-theme 3: Cognitive..."

Author response: This has been updated.

Discussion.

Expand with future lines of research.

Author response: This has been updated.

Conclusion.

This is a very limited section. The main findings of the study should be cited.

Author response: This has been updated.

References

The references section is not correct. References should be adapted to the journal's standards

Author response: This has been updated.

Round 2

Reviewer 2 Report

Dear Authors,

Thanks!

Kind regards

Author Response

Thank you for your feedback and support of this work. 

Reviewer 3 Report

Dear authors, I commend the authors for reviewing the manuscript. I suggest reviewing the English writing of the manuscript before considering it for publication.

Author Response

Thank you for your comments. We will review the manuscript for english and update.